# Brain disconnections link structural connectivity with function and behaviour

Michel Thiebaut de Schotten [ID] [1,2 ✉], Chris Foulon [ID] [3] & Parashkev Nachev[3]

Brain lesions do not just disable but also disconnect brain areas, which once deprived of their input or output, can no longer subserve behaviour and cognition. The role of white matter connections has remained an open question for the past 250 years. Based on 1333 stroke lesions, here we reveal the human Disconnectome and demonstrate its relationship to the functional segregation of the human brain. Results indicate that functional territories are not only defined by white matter connections, but also by the highly stereotyped spatial distribution of brain disconnections. While the former has granted us the possibility to map 590 functions on the white matter of the whole brain, the latter compels a revision of the taxonomy of brain functions. Overall, our freely available Atlas of White Matter Function will enable improved clinical-neuroanatomical predictions for brain lesion studies and provide a platform for explorations in the domain of cognition.

[1] Brain Connectivity and Behaviour Laboratory, Sorbonne University, Paris, France. [2] Groupe d'Imagerie Neurofonctionnelle, Institut des Maladies Neurodégénératives-UMR 5293, CNRS, CEA, University of Bordeaux, Bordeaux, France. [3] Institute of Neurology, UCL, London WC1N 3BG, UK. ✉email: michel.thiebaut@gmail.com

**Fig. 1 The biased distribution of ischaemic stroke. a** A representative brain lesion (left) together with its estimated disconnections (right). **b** The stroke lesion distribution (top row) compared with the synthetic lesion distribution (bottom row). **c** The two-dimension space visualisation of stroke (red) and synthetic lesion (blue) distribution. **d** The two-dimension space visualisation of stroke (red) and synthetic disconnectome (blue) distribution. See also Supplementary Fig. 1 for the exploration of a wide range of t-SNE parameters as well as uniform manifold approximation and projection for dimension reduction—UMAP[53]. Replication of the disconnection estimates in a lower resolution age matched sample of ten participants indicated a good reproducibility ($r = 0.866 \pm 0.066$).

Science relies on observation to elaborate theories and models of the nature and behaviour of natural things[1]. Observers notice pieces of evidence haphazardly, or manipulate their environment to produce new perceptible results. However, theories and models can be biased by the deceptive aspect of perception[2] and the incompleteness of data. For instance, unobservables cannot be detected, and are epistemically unavailable[3].

Neuroscience is no different, our understanding of function within the human brain originates from the observation of patients with focal brain lesions[4,5], particularly stroke, which amputates a part of the mind and impacts one person out of six[6]. Through patient observation, principal functions such as perception[7], emotion[8], memory[9], attention[10] and verbal communication[11] have been identified, localised and subsequently fractionated into more specific cognitive subprocesses. However, brain lesions, especially from ischaemic stroke, the most frequent kind[12], are not distributed randomly[13] (Fig. 1) and may have partially biased our taxonomy of brain functions.

In the past 20 years, following Meynert's original insight[14] our understanding of the functioning of the brain has evolved from a fractionated entity to an interconnected unity[15,16]. Brain zones considered critical to function have slowly altered over time, from those visibly damaged[17,18] to more distant, long-range zones that were disconnected[19–21]. Recent data even indicate that white matter disconnection might be a better predictor of brain dysfunction and recovery than the location of the lesion itself[22–24]. While our understanding of the anatomo-functional division of the brain surface is quite advanced[25,26], the relationship between brain disconnection and dysfunction remains limited and, accordingly, little is known about the functional division of white matter connections in the human brain[16].

Neuroimaging, particularly diffusion-weighted imaging (DWI) tractography, now enables the estimation of white matter disconnection after a brain lesion (i.e. disconnectome, Fig. 1a, ref. [27]). Joint models of lesions and their estimated disconnections both cohere with task-related activations from functional imaging and better explain clinical profiles[19,27–29]. However, whether the relationship between brain lesion and functional imaging findings is driven by biased clinical observation or the non-stochastic organisation of white matter connections in the human brain is unknown.

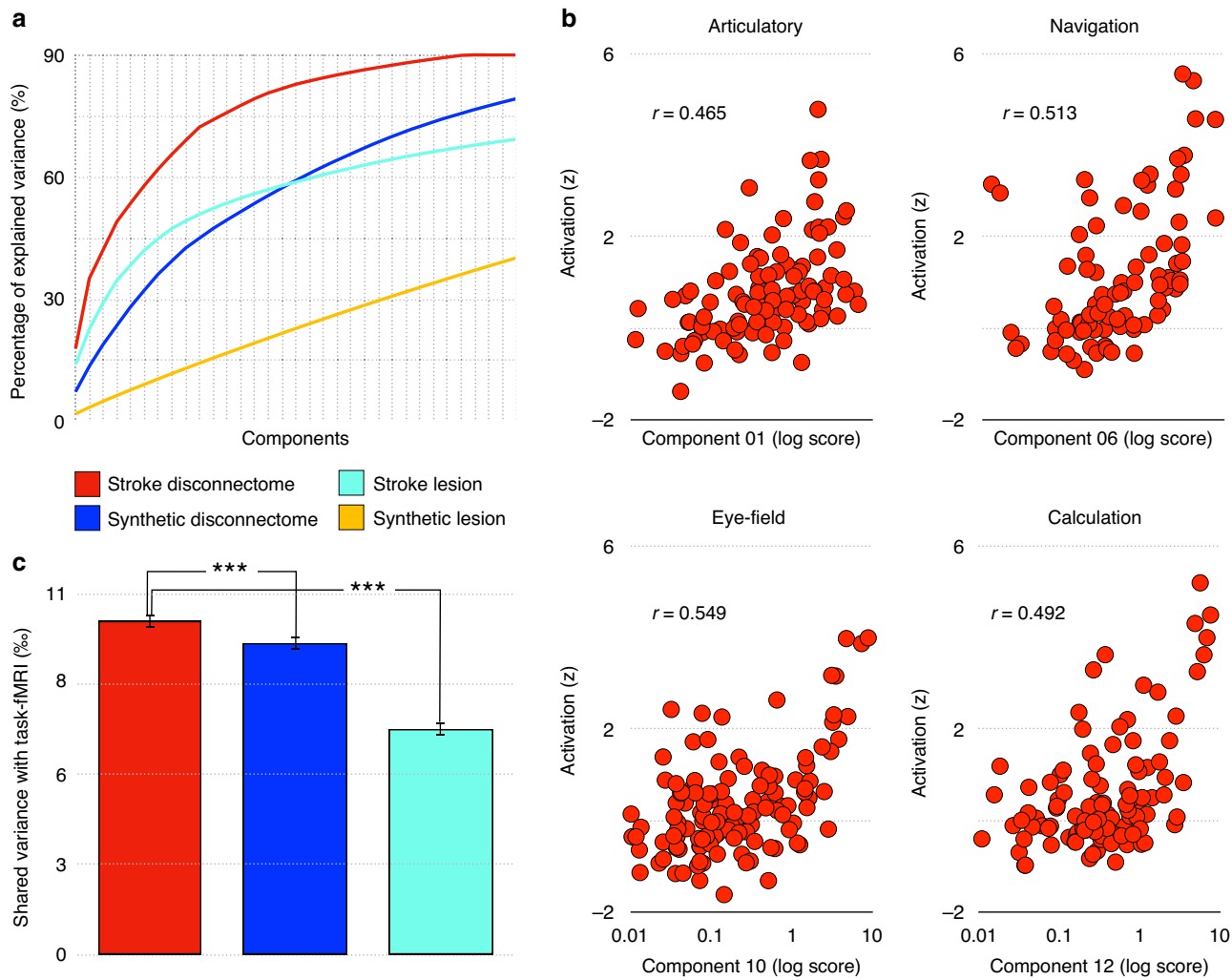

**Fig. 2 Principal components of stroke and their relationship to function. a** The cumulative percentage of explained variance for the first 30 components of the stroke disconnectome (red), synthetic disconnectome (blue), stroke lesion (turquoise), synthetic stroke lesion (yellow) distribution. By percentage of variance here we mean the $R^2$ between the original lesion or disconnectome and the lesion or disconnectome reconstructed from the linear combination of the components. **b** Four representative examples of the correlation between stroke disconnectome components and task-related functional MRI meta-analytic activations. **c** Average percentage of variance explained by task-related functional MRI meta-analytic activations (590 functions) for the stroke disconnectome (red, 46 components), synthetic disconnectome (blue, 53 components) and stroke lesion distribution (turquoise, 50 components). ***$p < 0.001$ two-tailed, bootstrapped ($n = 1000$) independent sample $t$-test (two comparisons, $p = 0.00099$ in both cases). Data are presented as mean values and error bars indicate 95% confidence intervals. Source data for this bar chart are available as Supplementary Data 2.

Based on the largest stroke collection ($n = 1333$)[30], combined with the most comprehensive database of neuroimaging meta-analysis[26] and the best white matter mapping derived from the Human Connectome Project 7T[31,32] we reveal the human disconnectome and demonstrate its relationship to the functional segregation of the human brain. Results indicate that functional territories are not only defined by white matter connections, but also by the highly stereotyped spatial distribution of brain disconnections. While the former has granted us the possibility to map 590 functions on the white matter of the whole brain, the latter compels a revision of the taxonomy of brain functions.

## Results

**The biased distribution of ischaemic stroke.** Taking advantage of an extensive set of 1333 real stroke lesions paired with a synthetic set of randomly distributed artificial lesions of the same size and lateralisation (Fig. 1b) we used a high-dimensional data non-linear embedding method (i.e. t-distributed stochastic neighbour

embedding, (t-SNE)[33]) to visualise the redundancy existing within the distribution of brain lesions (Fig. 1c) and their subsequent estimated disconnections (Fig. 1d). Lesions were placed in healthy connectomes to estimate disconnections[19,27,34–37]. The result indicates, as previously shown[13] that brain lesions show some redundancy and cluster together more than synthetic lesions (Fig. 1c). Noteworthy, strokes also seem to have a concentric distribution with a progressively increasing probability of damage from the surface of the brain to the deep white matter (a similar distribution can be found in refs. [13,23,38]). Importantly, disconnections in strokes (i.e. stroke disconnectome) demonstrated a higher level of clustering in comparison to the disconnection that was derived from paired synthetic lesions (i.e. synthetic disconnectome; Fig. 1d).

Redundancy in these datasets suggests that we should be able to summarise the pattern of brain areas disconnected by strokes into principal components.

In order to do so, the pattern of brain areas that were disconnected in each stroke was first characterised by measuring the average level of disconnection in subcortical areas[16] as well as

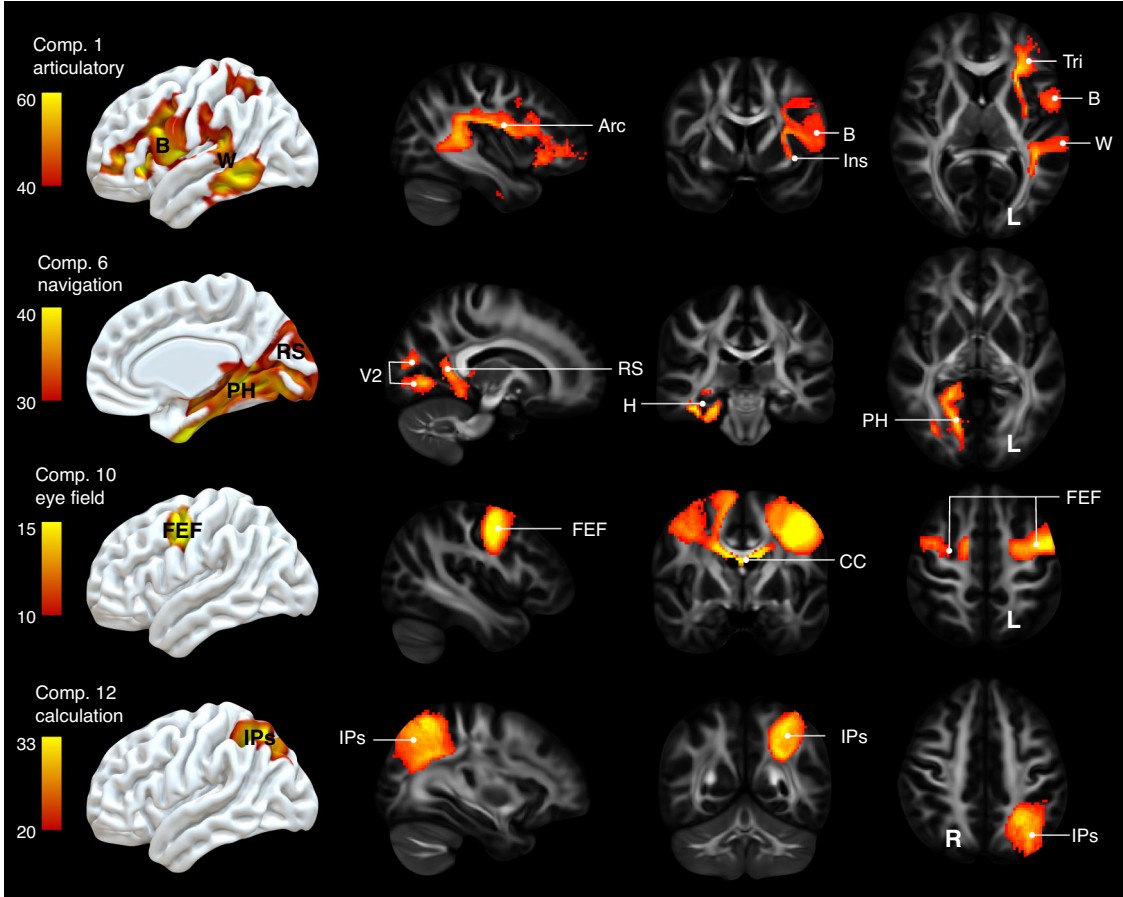

**Fig. 3 Functional mapping of white matter connections.** 3D representation of four representative components maps side-by-side with white matter sections. comp. component, Arc arcuate, B Broca area, Ins insula, Tri pars triangularis, W Wernicke, V2 secondary visual area, RS retrosplenial cortex, H hippocampus, PH parahippocampal area, FEF frontal eye field, CC corpus callosum, IPS intraparietal sulcus. Component maps were replicated a second time and indicated a good reproducibility (average Pearson $R = 0.813 \pm 0.079$).

in areas derived from a multimodal atlas of the brain surface[25]. Using a varimax-rotated principal component analysis (PCA) we modelled the main profiles of brain disconnections and revealed that 30 components, out of 46 in total, explained more than 90% of the variance of the regions disconnected by a stroke (Fig. 2a). All components were relatively independent from the lesion size (all $R^2 < 0.1$; see Supplementary Data 1 for a full list of the 46 components pattern of disconnection). In comparison, PCA of the direct lesion and the disconnection derived from the synthetic dataset only explained 70 and 80% of the variance with 30 components (Fig. 2a).

**Main components of stroke disconnection and their functions.** In order to assess the relationship between disconnection and brain function, the probability of disconnection of each component was compared to a manually curated version of the most extensive task-related functional magnetic resonance imaging (fMRI) meta-analytic dataset available[26,32]. Strikingly, 40 out of the 46 components disconnected a set of brain regions that significantly correlated with a set of specific task-related fMRI meta-analytic maps (Fig. 2b) with a small to large effect size (all $r > 0.202$, significant after Bonferroni correction for multiple comparisons—all $p < 0.00008$; see Supplementary Information B. for a full description of main correlations for each component and Supplementary Data 2 for a complete report of all correlations).

Whether the correspondence between the disconnectome components and task-related functional activations was due to the pure organisation of white matter in the brain or was influenced by the biased distribution of ischaemic strokes is unknown. To answer this question, we compared the correlation between task-related fMRI meta-analytic maps and all components of the stroke disconnectome, the synthetic disconnectome and brain lesions. Figure 2c shows that disconnections in strokes have a stronger relationship with task-related functional activations than the lesion alone ($t = 24.107$; $p < 0.001$, based on 1000 bootstrapped samples). Further, the correlation between task-related functional activations and the synthetic disconnectome have a significantly weaker relationship than with the stroke disconnectome ($t = 4.620$; $p < 0.001$, based on 1000 bootstrapped samples). These statistical differences suggest that on the one hand, the disconnectome corresponds with the underlying functional architecture (as revealed by fMRI) better than lesions alone. Hence brain disconnections are more appropriate to study the localisation of brain functions than brain lesions alone. On the other hand, our results also suggest that the non-random distribution of stroke has distorted the taxonomy of brain function underpinning the behavioural paradigms used in task-related fMRI and other brain mapping methods. In other words, the stereotyped location of stroke lesions has induced an observational bias in jointly impaired functions that, in turn, has biased the functional taxonomy used with fMRI in healthy subjects.

**Functional mapping of white matter connections.** Since most of the disconnectome components corresponded to task-related functional activations, we estimated the white matter structure

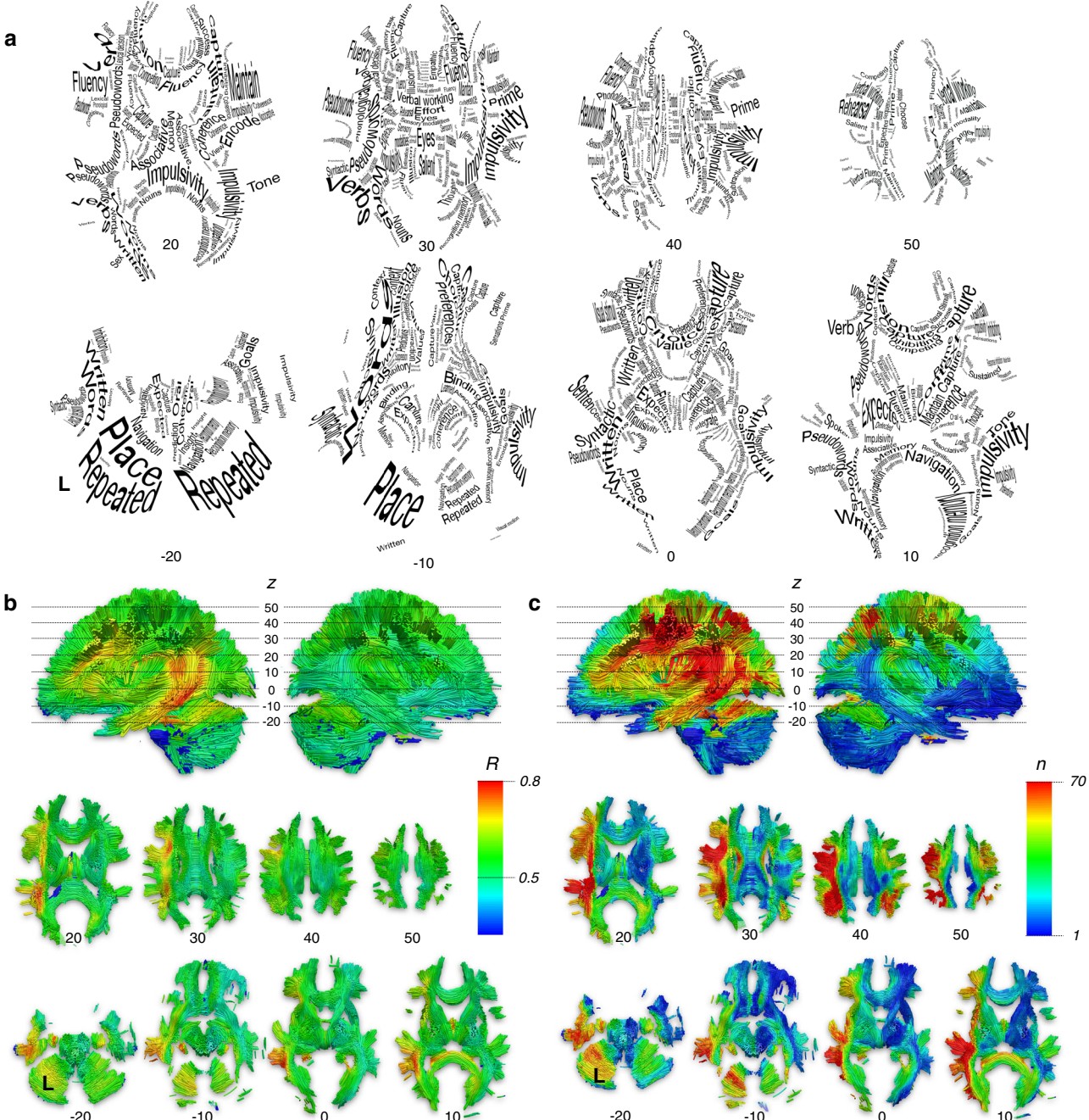

**Fig. 4 Atlas of white matter function. a** summary map of the white matter function only displaying terms with the highest statistical level (see Supplementary Figs. 50–57 for high resolution). **b** Effect size related to the prediction of white matter function. $R > 0.5$ indicates a large effect size. **c** Versatility maps. For each voxel of the MNI, the value indicates the number of terms related to the prediction of white matter function with a large effect size. The atlas white matter function was replicated a second time. The atlas of white matter function was replicated a second time and indicated a good reproducibility (average Pearson $R = 0.885 \pm 0.061$).

subjacent to each component allowing to derive a statistical map of the function of white matter connections [as disrupted by stroke]. Accordingly, component maps were calculated using permuted ($n = 1000$) linear regressions. In order to assess the reproducibility of the component maps, we split the dataset in two ($n = 666$). Pearson correlation indicated a good reproducibility across the two sets of component maps (average Pearson $R = 0.813 \pm 0.079$). As shown in Fig. 3, some components clearly demonstrated an intra-hemispheric network distribution, such as the articulatory loop that involves portions of the arcuate fasciculus and connects key territories including the Broca,

supramarginal and Wernicke areas[39]. Similarly, the navigation system involved the posterior portion of the cingulum that connects the right hippocampus[40] to the retrosplenial cortex[41,42]. Other components appeared to have a strong level of inter-hemispheric interaction, such as the left and right frontal eye fields that were bound via a portion of the corpus callosum tract. Such a strong interhemispheric association might explain why after a stroke[43] or unilateral electrical stimulation[44] of this functional field both eyes show an irrepressible deviation. Some components also had a more circumscribed distribution, such as calculation, which involved the left intraparietal sulcus and its

subjacent short white matter connections, as previously discussed in[45]. See Supplementary Information B. and Supplementary Figs. 3–48 for a full description of all the components, their link with functional neuroimaging and with symptoms when disconnected.

**Atlas of white matter function.** As some functions might emerge from the combination of the above components rather than a single component, a second level of permuted linear regression was performed to identify the statistical contribution of each voxel of each component map (i.e. independent variables) to each of the fMRI meta-analytic maps (i.e. the dependent variable). By doing so we obtained a white matter map that corresponded to each fMRI meta-analytic map. The same analysis was replicated in the second set of component maps. The two sets of maps indicated a good level of reproducibility (average Pearson $R = 0.885 \pm 0.061$). Results are summarised in Fig. 4a, which shows a comprehensive atlas of the function of white matter connections. For visualisation purposes, only the top contribution is represented here, however individual maps are available at neurovault.org (see 'Data availability' section). Mapping of the effect sizes (Fig. 4b) and the task versatility (Fig. 4c) indicated a strong asymmetry between the left and the right hemispheres. This asymmetry suggested that much less is known about the white matter function of the right hemisphere in comparison with the left. In addition, the component maps made available at neurovault.org (see 'Data availability' section) and have the potential to project any functional activation to the white matter and identify typical brain lesions leading to its disconnection (see Supplementary Information C. and Supplementary Fig. 49 for an example with left hand and right-hand finger tapping).

## Discussion

Applying state-of-the-art methods for synthesising meta-analytic functional mapping with white matter connectivity of the largest published set of acute stroke lesions we built an atlas of the function of white matter in the human brain.

As we could not directly collect high-resolution DWI from each patient due to limited clinical settings, instead, lesions were placed in healthy connectomes to estimate disconnections[19,27,34–37]. Importantly, we privileged the quality of the connectome rather than the age match when deriving the probability of disconnection induced by each lesion. While age-related changes in fractional anisotropy, number of streamlines/trajectories reconstructed and graph theoretical indices have been previously reported in the literature[46], none of these measures were used in our analyses. In the present case, age is not a confounding factor. As previously demonstrated, shape and spatial extent of average tracts[47], as well as disconnection estimates[27] are invariant across decades. In order to confirm this, the disconnection estimates were replicated in a lower resolution age matched sample of ten participants (i.e. Lifespan Human Connectome Project[48,49]) and indicated a good reproducibility ($r = 0.866 \pm 0.066$, see Supplementary Information E.).

The functions we localised in the atlas of the function of white matter correspond to the joint contribution of connected areas. This was made possible because of the redundancy in brain disconnection after a stroke that shows a striking correspondence with task-related fMRI activation patterns. Our result suggests that this correspondence is due to the influence of the organisation of white matter connections on the functional segregation of the human brain. Since the relationship between disconnection and task-related fMRI activation patterns was significantly stronger for real stroke lesion than simulated lesions, we suggest that the biased distribution of brain lesions has also biased our taxonomy of brain functions. While our method allowed us to map white matter function and may help to guide patients' symptoms exploration, the bias we report also provokes questioning and further investigation on the definition of brain functions, especially within the right hemisphere.

## Methods

The following workflow was summarised in Supplementary Fig. 2.

**Stroke lesions.** Lesion data were derived from 1333 patients admitted between 2001 and 2014 to University College London Hospitals (UCLH) with a clinical diagnosis of acute ischaemic stroke confirmed by DWI. Since DWI was routinely performed on the majority of attending patients, the sample was representative of the population, constrained mostly by contraindications and tolerability. The advantage of using a broad and unselected spectrum of strokes compared to a more narrowly defined cohort of stroke is that it is clinically relevant and generates characteristic patterns of disconnection that are the main variable of interest of our study. In contrast, narrowly defined cohorts of stroke still suffer from a biased distribution and are not representative of the stroke population and consequently cannot be used for data-driven clinical predictions. Age ranged from 18 to 97 years (mean 63.89, standard deviation 15.91), and the proportion of males was 0.561. The study was performed under ethical approval by the West London & GTAC Research Ethics Committee for consentless use of fully anonymised data. The majority of the data have been previously published[30]. The lesions used in this research were non-linearly registered to the Montreal neurological institute space ($2 \times 2 \times 2$ mm, MNI; http://www.bic.mni.mcgill.ca) so as to allow direct comparisons across individuals. Data were routinely acquired in the clinic and were anonymised prior to analysis and their collection was approved and monitored by the UK Health Research Authority. See ref. [30] for further details.

**Synthetic lesions.** To assess the impact of the spatial distribution of stroke lesion we computed synthetic lesions paired in size and hemispheric lateralisation with the stroke lesion dataset. To do so, we repeatedly divided each hemisphere of the brain mask of all the stroke lesions using $k$-mean clusterings[50] applied to the coordinates of each voxel varying the number of clusters from 2 to 50,000. This produced more than 1 million synthetic lesion masks. In order to minimise the bulkiness of the synthetic lesion, each mask was subsequently smoothed with a full-width half-maximum of 10 mm, thresholded at 0.3 and binarized. Finally, for each lesion of the stroke dataset, we sampled a lesion of the synthetic pool with the same size and localised in the same hemisphere. This produced a dataset of 1333 synthetic lesions exactly paired with the stroke dataset but pseudo-randomly distributed in the brain. The code used for the production of synthetic lesion paired with real lesions is available as supplementary code (see 'Code availability' section)

**Disconnectome.** Similarly to previous work[19,27,35] the probability of disconnection induced by each lesion was computed with the 'disconnectome map' tool of the BCBtoolkit software[27].

Firstly, we tracked white matter fibres that pass through each lesion in 163 healthy controls tractographies (0.448 males) dataset acquired at 7 T by the Human Connectome Project Team[31] and computed according to[32] (see Supplementary Information A. for full details on the processing of the DWI data). For each lesion, tractography maps of the 163 healthy controls were subsequently binarised and averaged together so that each voxel represented a probability of disconnection from 0 to 1. This produced a stroke lesion disconnectome dataset ($n = 1333$) and a synthetic lesion disconnectome dataset ($n = 1333$).

**Spatial embedding.** Stroke lesions and synthetic lesions, as well as their respective disconnectome maps' spatial distributions, were visually compared using a machine learning visualisation approach, t-SNE[33]. t-SNE provided a two-dimensional visualisation of the similarity and difference between the maps so that a smaller distance between two points represented a higher similarity between the maps. t-SNE was computed with the help of NiBabel 3.1.1, scikit learn 0.23 in Python 3 with a perplexity of 30, an exaggeration of 12 (to increase the space between clusters) a learning rate of 200 and 1000 iterations. This step provided us with the visualisation of the stroke and synthetic lesions and disconnectome map distribution that is displayed in Fig. 1a, b.

**Data compression.** In order to properly deal with redundancy, lesions and disconnectome of the stroke and the synthetic datasets were first characterised using the multimodal parcellation of human cerebral cortex (MMP, ref. [25]) and subsequently entered a varimax-rotated PCA.

The MMP provided 360 cortical areas very well characterised by their anatomy and functional specificity. As subcortical areas also play an important role in cognition, we added 12 additional regions, defined manually, including the amygdala, the caudate nucleus, the hippocampus, the pallidum, the putamen and the thalamus for the left and the right hemispheres, respectively.

For each lesion, the proportion of damage (for the lesions) or the probability of disconnection (for the disconnectome) was estimated for each region of interest. This step produced four matrices in total—2 (lesion or disconnectome) × 2 (stroke or synthetic).

Each matrix entered a PCA using a covariance matrix and varimax rotation (with a maximum of 500 iterations for convergence) in order to estimate the number of principal components to extract for the four conditions (Fig. 1). Plot of the components according to their cumulative explained variance can be seen in Fig. 2a. Component scores were systematically extracted for all components identified by the PCA by means of multiple regression. This provided us with the contribution of each parcel of the MMP and subcortical areas to each component.

**Relationship to task-related fMRI metanalysis**. We used a meta-analytic database[26] (http://Neurosynth.org) that summarises the details of 11,406 fMRI literature sources and manually curated it as described in ref. [32]. The manual curation consisted in the previously published selection of 590 maps related to specific cognitive processes out of the whole Neurosynth database.

The curated database represented 590 cognitive term maps that were converted into a matrix using the average z value for each MMP parcels and the manually defined subcortical areas mentioned above.

We estimated the relationship between each term of the task-related fMRI metanalysis matrix and the component scores extracted in the previous section using Pearson correlation.

Comparison between the stroke lesion, disconnectome and synthetic disconnectome relationship with task-related fMRI metanalysis was assessed by means of a bootstrapped ($n = 1000$) independent sample t-test.

**Components maps**. We used a permuted ($n = 1000$) multiple regression to statistically assess the relationship between each component and voxel in the white matter. To do so, we used the component score as an independent variable and the voxel probability of disconnection in the stroke disconnectome maps as dependent variables. This analysis was made possible thanks to the function *Randomise* as part of the software package FSL (https://fsl.fmrib.ox.ac.uk/fsl)[51,52].

To address the replicability of our maps, the 1333 disconnectome maps dataset was split in two 666 datasets and the multiple regression was run twice. Quality of the results duplication was assessed by means of Pearson correlations between the two set of maps derived from the analyses. This section resulted in two white matter maps per component.

**Atlas of the white matter function**. Next, we explored the contribution of each component map voxels to the task-related fMRI meta-analytic maps. A permuted ($n = 1000$) linear regression was computed between the correlation value of each task-related fMRI meta-analytic map as an independent variable and each component maps voxels as a dependant variable. This analysis was run twice, once for each set of the component maps and its replication quality was assessed as previously mentioned. This section resulted in two white matter maps per task-related fMRI meta-analytic term.

A summary map of the white matter function was computed displaying only the map with the highest statistical level (*find_the_biggest* in FSL). An R (i.e. goodness of fit) was calculated for each voxel in order to provide a visualisation of the effect size of the summary map. As several functions can load on the same tract, we assessed versatility by counting the number of tasks having an effect size $R > 0.3$ for each voxel.

**Visualisation**. A visualisation of the results was performed using Surf Ice https://www.nitrc.org/projects/surfice/ and Trackvis http://trackvis.org.

**Reporting summary**. Further information on research design is available in the Nature Research Reporting Summary linked to this article.

## Data availability
The two sets of component maps and the atlas of white matter function (original and replication) are available at https://identifiers.org/neurovault.collection:7735. The atlas of white matter function is also available at A–C terms: https://identifiers.org/neurovault.collection:7756. D–H terms: https://identifiers.org/neurovault.collection:7757. I–N terms: https://identifiers.org/neurovault.collection:7758. O–R terms: https://identifiers.org/neurovault.collection:7759. S–U terms: https://identifiers.org/neurovault.collection:7760. V–Z terms: https://identifiers.org/neurovault.collection:7761. The raw dataset analysed in the current study is available at https://www.humanconnectome.org (7 T diffusion data) and http://www.neurosynth.org (metanalytic functional MRI maps). In addition, processed data are available at http://www.bcblab.com/BCB/Opendata.html and https://osf.io/5zqwg/ and on request to the corresponding author.

## Code availability
The code used in the analyses is available as part of the BCBtoolkit package http://toolkit.bcblab.com. The code used for the production of paired synthetic lesions as well as the effect size maps is available at https://github.com/chrisfoulon/BCBlib/blob/devel/bcblib/

scripts/generate_synth_lesions.py, https://github.com/chrisfoulon/BCBlib/blob/devel/bcblib/scripts/pick_up_matched_synth_lesions.py, and https://github.com/chrisfoulon/BCBlib/blob/devel/bcblib/scripts/effectsize_T2R.py. Any additional information is available on request to M.T.d.S.

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

## Acknowledgements

We thank Tianbo Xu, who generated all the stroke lesion masks, and Lauren Sakuma, for useful discussion and edits to the manuscript. We thank Maurizio Corbetta, Stephanie Forkel, Emmanuelle Volle, Patrick Friedrich and Sandrine Cremona for discussion of the results. We also thank Laurent Petit and his team (GIN) for providing us with *f*MRI maps of left and right hands finger tapping as well as University of Bordeaux and CNRS for the infrastructural support. This project has received funding from the European Research Council (ERC) under the European Union's Horizon 2020 research and innovation programme (grant agreement no. 818521 to M.T.d.S.). P.N. is funded by the Wellcome Trust and the UCLH NIHR Biomedical Research Centre.

## Author contributions

M.T.d.S. conceived and coordinated the study, implemented the methods, performed the analyses, wrote the manuscript and provided funding. C.F. implemented the methods and wrote the manuscript. P.N. conceived and coordinated the study, collected and reviewed the neuroimaging data, wrote the manuscript and provided funding.

## Competing interests

The authors declare no competing interests.
