## [Peer Review File · Nature Communications]

Reviewers' Comments:

Reviewer #1:

Remarks to the Author:

This is a very impressive study. The authors document the effect of lesions on brain disconnection by analysing the impact of more than 1300 strokes. The work is likely to have substantial scientific and clinical impact, given our emerging understanding that lesion-symptom relationships are not well understood by examining damage to grey matter alone – and yet this assumption is still central to the way we interpret many neuropsychological studies. Although we have had detailed knowledge of which areas of the brain are most vulnerable to ischemic stroke for many years, the impact of this pattern on disconnections has not been well-characterised. This study uses machine learning and PCA to characterise this non-randomness and this approach seems very powerful.

I only have a few comments:

1. It would be helpful if the lesion characteristics retained in the synthetic lesions could be made more explicit and if their characteristics could be better justified. The methods mention that synthetic lesions are paired in size and hemisphere – what about shape and grey/white matter ratio – are they relevant? Is there any need to pay attention to the type of ischemic stroke – e.g. thrombotic vs. embolic strokes? Large vs. small vessel stroke? I wonder if the intention is to produce a synthetic control sample that does not retain these characteristics of real strokes, since if they were matched, the synthetic strokes might not look different from real strokes -- but articulating this would help readers to understand what the overarching aim is.
2. In Figure 2 (PCA analysis), I lost track of what variance was being explained (please clarify in the footnote for Fig. 2a). For this reason, it wasn't clear to me why synthetic disconnections would explain more variance than real lesions as the number of components becomes large. Would it be useful to add synthetic lesions alongside stroke lesions, so we can see if the additional variance explained is greater for real vs. synthetic disconnections than real vs. synthetic lesions? How many components do the various types of data suggest would be an appropriate number?
3. I think the statement "the non-random distribution of stroke has distorted the taxonomy of brain function in task fMRI" needs some unpacking. I couldn't directly see the evidence for this in Figure 2c.
4. In the labelling of the components in Fig 3 and 4 (and in the supplementary materials), it would be useful if more than a single cognitive or task term could be used to summarise function. These terms are derived from Neurosynth and there will be a whole field of terms that relate, with different strengths, to each connection component. It would be helpful to present this whole field, at least in the supplementary materials, given that each connection component is almost certainly involved in multiple tasks or aspects of cognition.

Reviewer #2:

Remarks to the Author:

This paper presents an analysis of an extensive dataset of stroke lesions, in order to characterize the "disconnectome" of the human brain. Although there is a long history of meta-analysis in neuroimaging and some previous work applying these methods to lesion data, I think it's fair to say that there has been nothing even close to this kind of analysis published before. The visualization in Figure 4a is particularly amazing. I think it's a very important advance, but I do have a few comments that I hope will help improve the manuscript.

My only conceptual question is the degree to which the observed components relate to cerebrovascular territories. Is there some way to assess what the analyses tell us beyond the fact that strokes happen in consistent locations due to consistent vasculature?

Minor comments:

1. page 3: the claim that "our understanding of function within the human brain is largely based upon on the observation of patients with focal brain lesions" was true 50 years ago but seems a bit overblown today, given how much we have learned from neuroimaging of healthy individuals over the last few decades.
2. I have two comments about the presentation of the t-SNE results. First, it's a bit odd to call this a "machine learning algorithm" - it's no more so than PCA or any other dimensionality reduction method. I would call it a "dimensionality reduction method" instead. Second, t-SNE results can be notoriously variable - do the results hold up across multiple runs of the algorithm?
3. Figure 2a is very difficult to parse - I would suggest a line graph rather than overlapping colored bars.
4. It would be useful to label all image slices with L/R since different subfields vary in the L/R orientation in which data are presented.
5. It would be useful for the component maps and atlas maps to be uploaded to a data sharing repository where they can be viewed in detail, such as Neurovault.
6. It would be great if the raw lesion maps could also be shared openly.

Signed,
Russ Poldrack

Reviewer #3:

Remarks to the Author:

Many thanks for giving me the opportunity to review this paper. Here, the authors studied patients with stroke lesions to assess disconnection between different brain areas, and to explore functional associations. The paper was overall creative and benefitted from inclusion of a large stroke database. On the other hand, I missed important details on the patient population and also a cross-validation of findings based on patient specific connectome information. Further, I was not clear on the quantitative analyses steps performed, specifically wrt the synthetic null models and statistical inferences supporting the conclusions. The significance of the neurosynth associations was not fully clear to me. Please find my detailed comments below:

1) Figure 1 shows that the actual stroke lesion distribution is more clustered than a synthetic lesion distribution, and lesions as well as disconnections are shown at more extreme locations in tSNE derived embedding space. While this approach is interesting and the visualization elegant, I have two questions:

- first, I am wondering whether the authors' synthetic null model is a good one here. From the methods, it was not entirely clear to me whether the model actually controls adequately for a) typical 'autocorrelation' in the brain eg with respect to structural but also connectivity related features (which should generally lead to higher 'clustering') and whether b) vascular territory (which one would assume biases the location of stroke lesions in the brain).

- second, the statistical testing carried out to support the conclusions in Figure 1c and d are missing in the main description, and would be helpful to add. Specifically, shouldn't a testing against synthetic null models be generally based on something like $p < 0.05$ thresholds, similar to typical permutation based null models for other questions? If so, while it's evident that the lesions have higher eccentricity in embedding space than the synthetic null lesions models, the findings do not generally support a

95% consistency in such a measure (at least visually). Are there some metrics that can be calculated to confirm that real lesions are indeed further outside in embedding space than 95% of synthetic lesions?

2) The initial description of the methodology in the main text (line 79-92) does not clearly reveal whether lesions were placed in a healthy connectome (which is what the authors ultimately did), or whether diffusion MRI connectivity changes were directly measured by comparing diffusion MRI data in patients relative to controls. I suggest to be more clear on this part in the main part of the paper, even already in the abstract and introduction.

3) One question that naturally occurs then is whether the HCP (young adult) dataset connectome is representative to the brain of stroke patients, which are suspected to be older and potentially more atrophic generally and suffering from pre-existing conditions. A validation of the findings based on connectome data from stroke patients could be useful to mitigate some of these concerns.

4) I could not find any socio-demographic or clinical details on the stroke population in the current paper, nor a description of how lesions were defined. There was a reference to Ref 38, which however also did not provide extensive details on overall clinical and socio-demographic aspects of the stroke patients.

5) I am not fully clear about the findings shown in Figure 2b. The authors discovered 30 different components (30) and correlated those with neurosynth term maps. Given that the term base in neurosynth seems quite large (I think its currently more than 1k), it is not surprising to obtain several significant associations here. Are corrections for multiple comparisons carried out in this analysis?

Minor comments:

- Can the authors speculate whether the need for a lower number of PCA components (see lines 112-119) naturally follows from a less clustered arrangement in t-SNE space (see Fig 1). I would have generally expected this, but curious to hear the authors thoughts and experiences.

- In line 123, Neurosynth appears to be referred to as a manually curated meta-analytic dataset. Can the authors clarify this? It was my understanding that neurosynth is based on automated abstract parsing.

- I'd recommend more details on the structural connectome generation (wrt tractography etc). Its currently hard to know which precise methods were used for preprocessing, tract tracing, and averaging of connectomes across subjects.

REVIEWER COMMENTS

We want to thank the reviewers for their very positive and encouraging comments. We also thank the reviewers for their suggestions, which significantly improved the manuscript. Changes have been highlighted in cyan in the main text and the supplementary material. We also systematically reported the manuscript changes in our point by point response below.

Reviewer #1 (Remarks to the Author):

This is a very impressive study. The authors document the effect of lesions on brain disconnection by analysing the impact of more than 1300 strokes. The work is likely to have substantial scientific and clinical impact, given our emerging understanding that lesion-symptom relationships are not well understood by examining damage to grey matter alone – and yet this assumption is still central to the way we interpret many neuropsychological studies. Although we have had detailed knowledge of which areas of the brain are most vulnerable to ischemic stroke for many years, the impact of this pattern on disconnections has not been well-characterised. This study uses machine learning and PCA to characterise this non-randomness and this approach seems very powerful.

Thank you for your very positive feedback.

I only have a few comments:

1. It would be helpful if the lesion characteristics retained in the synthetic lesions could be made more explicit and if their characteristics could be better justified. The methods mention that synthetic lesions are paired in size and hemisphere – what about shape and grey/white matter ratio – are they relevant? Is there any need to pay attention to the type of ischemic stroke – e.g. thrombotic vs. embolic strokes? Large vs. small vessel stroke? I wonder if the intention is to produce a synthetic control sample that does not retain these characteristics of real strokes, since if they were matched, the synthetic strokes might not look different from real strokes -- but articulating this would help readers to understand what the overarching aim is.

This is an unselected series of strokes that will include thrombotic and embolic strokes. The advantage of being unselected is that they are reasonably representative of the population. We added the following information in the maintext.

“Lesion data was derived from 1333 patients admitted between 2001 and 2014 to University College London Hospitals (UCLH) with a clinical diagnosis of acute ischaemic stroke confirmed by diffusion weighted imaging (DWI). Since DWI was routinely performed on the majority of attending patients, the sample was representative of the population, constrained mostly by contraindications and tolerability. Age ranged from 18 to 97 years (mean 63.89, standard deviation 15.91), and the proportion of males was 0.561. The study was performed under ethical approval by the local research ethics committee for consentless use of fully anonymized data. The majority of the data has been previously published in Xu et al. (2018).”

2. In Figure 2 (PCA analysis), I lost track of what variance was being explained (please clarify in the footnote for Fig. 2a). For this reason, it wasn't clear to me why synthetic disconnections would explain more variance than real lesions as the number of components becomes large. Would it be useful to add synthetic lesions alongside stroke lesions, so we can see if the additional variance explained is greater for real vs. synthetic disconnections than real vs. synthetic lesions? How many components do the various types of data suggest would be an appropriate number?

The number of components chosen is subjective, the more components chosen the higher the percentage of variance represented. Here we displayed 33 components that correspond to 90% of the variance explained for the disconnectome of stroke lesions. By percentage of variance explained we mean the R^2 between the original lesion and or disconnectome and the lesion or disconnectome reconstructed from the linear combination of the components.

We now added the synthetic lesions in Fig 2a and clarified in the caption what we mean as 'explained variance'

“the cumulative percentage of explained variance for the first 30 components of the stroke disconnectome (red), synthetic disconnectome (blue), stroke lesion (green), synthetic stroke lesion (yellow) distribution. By percentage of variance here we mean the R^2 between the original lesion and or disconnectome and the lesion or disconnectome reconstructed from the linear combination of the components”

3. I think the statement “the non-random distribution of stroke has distorted the taxonomy of brain function in task fMRI” needs some unpacking. I couldn't directly see the evidence for this in Figure 2c.

We now unpacked this statement as follows

“These statistical differences suggest that on the one hand, the disconnectome corresponds with the underlying functional architecture (as revealed by fMRI) better than lesions alone. Hence brain disconnections are more appropriate to study the localisation of brain functions than brain lesions alone. On the other hand, our results also suggest that the non-random distribution of stroke has distorted the taxonomy of brain function underpinning the

behavioural paradigms used in task-related fMRI and other brain mapping methods. In other words, the stereotyped location of stroke lesions has induced an observational bias in jointly impaired functions that, in turn, has biased the functional taxonomy used with fMRI in healthy subjects.”

4. In the labelling of the components in Fig 3 and 4 (and in the supplementary materials), it would be useful if more than a single cognitive or task term could be used to summarise function. These terms are derived from Neurosynth and there will be a whole field of terms that relate, with different strengths, to each connection component. It would be helpful to present this whole field, at least in the supplementary materials, given that each connection component is almost certainly involved in multiple tasks or aspects of cognition

This is an important point. We now added a supplementary table 2 that includes all correlations for each component.

This information has been added in the main text accordingly.

“(all $r > 0.202$, significant after Bonferroni correction for multiple comparisons— all $p < 0.00008$; see supplementary material for a full description of main correlations for each component and supplementary table 2 for a complete report of all correlations).”

Reviewer #2 (Remarks to the Author):

This paper presents an analysis of an extensive dataset of stroke lesions, in order to characterize the “disconnectome” of the human brain. Although there is a long history of meta-analysis in neuroimaging and some previous work applying these methods to lesion data, I think it’s fair to say that there has been nothing even close to this kind of analysis published before. The visualization in Figure 4a is particularly amazing. I think it’s a very important advance, but I do have a few comments that I hope will help improve the manuscript.

Thank you! We’re very grateful for your highly positive feedback.

My only conceptual question is the degree to which the observed components relate to cerebrovascular territories. Is there some way to assess what the analyses tell us beyond the fact that strokes happen in consistent locations due to consistent vasculature?

The structure of vascular damage imposes a structure downstream on the patterns of functional deficits that reflects the intersection between the vascular patterns and the underlying functional organisation -- both white matter and grey matter. Our aim is to clarify the interaction between the vascular structure and the white matter structure in a way that reveals commonalities in the observed patterns of functional deficits.

Additionally to the fact that strokes happen in consistent locations due to consistent vasculature we can see that strokes typically have a concentric distribution with a

progressively increasing probability of damage from the surface of the brain to the deep white matter.

This information has been added in the text

“Noteworthy, strokes also seems to have a concentric distribution with a progressively increasing probability of damage from the surface of the brain to the deep white matter (a similar distribution can be found in Husain and Nachev 2006; Mah et al. 2014 and Corbetta et al. Neuron 2015)”

Minor comments:

1. page 3: the claim that "our understanding of function within the human brain is largely based upon on the observation of patients with focal brain lesions" was true 50 years ago but seems a bit overblown today, given how much we have learned from neuroimaging of healthy individuals over the last few decades.

Thank you we rephrased this claim accordingly

“our understanding of function within the human brain originates from the observation of patients with focal brain lesions”

2. I have two comments about the presentation of the t-SNE results. First, it’s a bit odd to call this a “machine learning algorithm” - it’s no more so than PCA or any other dimensionality reduction method. I would call it a “dimensionality reduction method” instead. Second, t-SNE results can be notoriously variable - do the results hold up across multiple runs of the algorithm?

We changed the text accordingly for

“we used a high-dimensional data non linear embedding method (i.e. T-distributed Stochastic Neighbor Embedding, T-SNE, van der Maaten & Hinton, 2008) to visualise the redundancy existing within the distribution of brain lesions (Figure 1c) and their subsequent estimated disconnections (Figure 1d).”

We have explored a wide range of tSNE parameters with no substantial impact on the final pattern and have replicated essentially the same result with umap (i.e.:brain lesions show some redundancy and cluster together more than synthetic lesions and disconnections in strokes demonstrated a higher level of clustering in comparison to the disconnection that was derived from paired synthetic lesions) now in supplementary material.

Supplementary figure 1: Replication of The biased distribution of ischemic stroke with TSNE (a,b: perplexity = 40; c,d: perplexity = 60; e,f: perplexity = 80) and U-map. Left panel corresponds to the two-dimension space visualisation of stroke (red) and synthetic lesion (blue) distribution. Right panel corresponds to the two-dimension space visualisation of stroke (red) and synthetic disconnectome (blue) distribution.

3. Figure 2a is very difficult to parse - I would suggest a line graph rather than overlapping colored bars.

We modified the figure accordingly.

4. It would be useful to label all image slices with L/R since different subfields vary in the L/R orientation in which data are presented.

Done

5. It would be useful for the component maps and atlas maps to be uploaded to a data sharing repository where they can be viewed in detail, such as Neurovault.

The component maps and atlas maps are already available as supplementary material.

Additionally following your recommendation we uploaded these maps on Neurovault

The component maps can be found here <https://identifiers.org/neurovault.collection:7735>

The atlas of white matter function here

A to C terms: <https://identifiers.org/neurovault.collection:7756>

D to H terms: <https://identifiers.org/neurovault.collection:7757>

I to N terms: <https://identifiers.org/neurovault.collection:7758>

O to R terms: <https://identifiers.org/neurovault.collection:7759>

S to U terms: <https://identifiers.org/neurovault.collection:7760>

V to Z terms: <https://identifiers.org/neurovault.collection:7761>

These links have been added to the manuscript

6. It would be great if the raw lesion maps could also be shared openly.

The regulatory terms under which the lesion maps are made available to researchers are that they should be supplied on request rather than published on a public website.

Signed,

Russ Poldrack

Reviewer #3 (Remarks to the Author):

Many thanks for giving me the opportunity to review this paper. Here, the authors studied patients with stroke lesions to assess disconnection between different brain areas, and to explore functional associations. The paper was overall creative and benefitted from inclusion of a large stroke database. On the other hand, I missed

important details on the patient population and also a cross-validation of findings based on patient specific connectome information. Further, I was not clear on the quantitative analyses steps performed, specifically wrt the synthetic null models and statistical inferences supporting the conclusions. The significance of the neurosynth associations was not fully clear to me. Please find my detailed comments below:

1) Figure 1 shows that the actual stroke lesion distribution is more clustered than a synthetic lesion distribution, and lesions as well disconnections are shown at more extreme locations in tSNE derived embedding space. While this approach is interesting and the visualization elegant, I have two questions:

- first, I am wondering whether the authors' synthetic null model is a good one here. From the methods, it was not entirely clear to me whether the model actually controls adequately for a) typical 'autocorrelation' in the brain eg with respect to structural but also connectivity related features (which should generally lead to higher 'clustering') and whether b) vascular territory (which one would assume biases the location of stroke lesions in the brain).

The null model seeks to represent a set of lesions of comparable volume and central spatial location but a pattern of covariance of detailed spatial features determined by the intersection of a sphere with the parenchyma of the brain, i.e. to approximate a real lesion in every way *other* than vascular anatomy and the way in which it interacts with occlusion and stenosis to produce ischaemic lesions. This is because our main model is addressed to the question of how the vascular anatomy interacts with white matter tracts to generate characteristic patterns of disconnection we are here seeking to map definitively.

In order to help with the replication and extension of our findings we now also provide the code that produced the matched synthetic lesion that can be applied to any other datasets.

- second, the statistical testing carried out to support the conclusions in Figure 1c and d are missing in the main description, and would be helpful to add. Specifically, shouldn't a testing against synthetic null models be generally based on something like $p < 0.05$ thresholds, similar to typical permutation based null models for other questions? If so, while its evident that the lesions have higher eccentricity in embedding space than the synthetic null lesions models, the findings do not generally support a 95% consistency in such a measure (at least visually). Are there some metrics that can be calculated to confirm that real lesions are indeed further outside in embedding space than 95% of synthetic lesions?

There is no established statistical test for the heterogeneity of a non-linear latent embedding: the purpose of Figure 1 is to demonstrate visually the differences in the latent structure that are formalised in subsequent analyses.

2) The initial description of the methodology in the main text (line 79-92) does not clearly reveal whether lesions were placed in a healthy connectome (which is what the authors ultimately did), or whether diffusion MRI connectivity changes were directly measured by comparing diffusion MRI data in patients relative to controls. I suggest to be more clear on this part in the main part of the paper, even already in the abstract and introduction.

It is essentially impossible to obtain high quality tractographic data from a stroke population owing to the limited tolerance of the long scanning times involved. It is true that some differences are to be expected but the broad organisation of the white matter tracts that is the driver of the disconnectome can be reasonably expected to be preserved.

We clarified the text accordingly

“Taking advantage of an extensive set of 1,333 real stroke lesions paired with a synthetic set of randomly distributed artificial lesions of the same size and lateralisation (Figure 1b) we used a high-dimensional data non-linear embedding method (i.e. T-distributed Stochastic Neighbor Embedding, T-SNE, ²⁹) to visualise the redundancy existing within the distribution of brain lesions (Figure 1c) and their subsequent estimated disconnections (Figure 1d). Lesions were placed in healthy disconnectomes to estimate disconnections^{19, 26, 30, 31, 32, 33}. The result indicates, as previously shown¹³ that brain lesions show some redundancy and cluster together more than synthetic lesions (Figure 1c).”

3) One question that naturally occurs then is whether the HCP (young adult) dataset connectome is representative to the brain of stroke patients, which are suspected to be older and potentially more atrophic generally and suffering from pre-existing conditions. A validation of the findings based on connectome data from stroke patients could be useful to mitigate some of these concerns.

This is a good point. We previously demonstrated that there is no effect of age on the shape and trajectory of white matter tracts (supplementary material of Rojkova et al BSAF 2016). We also previously explored in the supplementary material of Foulon et al. 2018 the effect of age on the disconnectome and found no difference. We reproduced below the previously published analysis in full for your assessment.

In sum, there is no effect of age on the macrostructural anatomy of the disconnectome.

The optimal number of participants was calculated for disconnectome maps from separate paired populations of equal gender distribution. This approach was repeated for groups consisting of 4, 6, 8, 10, 12, 14, 16, 18 and 20 subjects. Squared spatial Pearson's correlations between each pair (i.e. square of fsfcc from FSL) were employed to calculate the percentage of shared variance (i.e. the similarity). This analysis indicates a steep increase of shared variance between disconnectome maps produced from 4 to 10 participants followed by a slower increase from 10 to 20 participants. This result indicates that, using the disconnectome, 10 subjects are sufficient to produce a good enough disconnectome map that matches the overall population (above 70% of shared variance). A larger dataset (n = 36) can be downloaded on our website (<http://www.bcblab.com/opendata>). Additionally, HCP 7T data (n = 166) have been prepared for the disconnectome and are available on demand to the authors (hd.chrisfoulon@gmail.com or michel.thiebaut@gmail.com).

We also measured whether the shape of the disconnectome changes over age. We assessed this question by producing disconnectome maps for each decade. We quantified similarities using squared spatial Pearson's for the 21-30-year-old maps and the maps for the other decades. The result indicates that disconnectome maps show a very high anatomical similarity between decades. Hence disconnectome maps in our sample did not show any age-related changes.

4) I could not find any socio-demographic or clinical details on the stroke population in the current paper, nor a description of how lesions were defined. There was a reference to Ref 38, which however also did not provide extensive details on overall clinical and socio-demographic aspects of the stroke patients.

This is an unselected series of strokes that will include thrombotic and embolic strokes. The advantage of being unselected is that they are reasonably representative of the population. We added the following information in the maintext.

“Lesion data was derived from 1333 patients admitted between 2001 and 2014 to University College London Hospitals (UCLH) with a clinical diagnosis of acute ischaemic stroke confirmed by diffusion weighted imaging (DWI). Since DWI was routinely performed on the majority of attending patients, the sample was representative of the population, constrained mostly by contraindications and tolerability. Age ranged from 18 to 97 years (mean 63.89, standard deviation 15.91), and the proportion of males was 0.561. The study was performed under ethical approval by the local research ethics committee for consentless use of fully anonymized data. The majority of the data has been previously published In Xu et al., 2017”

5) I am not fully clear about the findings shown in Figure 2b. The authors discovered 30 different components (30) and correlated those with neurosynth term maps. Given that the term base in neurosynth seems quite large (I think its currently more than 1k), it is not surprising to obtain several significant associations here. Are corrections for multiple comparisons carried out in this analysis?

Yes we did correct for multiple comparisons and have now clarified this point in the manuscript.

“In order to assess the relationship between disconnection and brain function, the probability of disconnection of each component was compared to a manually curated version of the most extensive fMRI meta-analytic dataset available (25). The manual curation consisted in the previously published selection of 590 maps related to specific cognitive processes out of the whole Neurosynth database (30). Strikingly, 40 out of the 46 components disconnected

a set of brain regions that significantly correlated with a set of specific task-related fMRI meta-analytic maps (Figure 2b) with a small to large effect size (all $r > 0.202$, significant after Bonferroni correction for multiple comparisons— all $p < 0.00008$; see supplementary material for a full description of main correlations for each component and supplementary table 2 for a complete report of all correlations)."

Minor comments:

- Can the authors speculate whether the need for a lower number of PCA components (see lines 112-119) naturally follows from a less clustered arrangement in t-SNE space (see Fig 1). I would have generally expected this, but curious to hear the authors thoughts and experiences.

It is rather hard to infer the number of linear components that could be extracted looking at the non-linear embedding of the data.

What we observed in our dataset was that the apparently more clustered data in the non-linear embedding, the higher the redundancy in the dataset and the lower the number of components needed to capture the variance of the dataset. We think this is because the tighter the adherence to whatever structure we observe the lower the number of components likely required to capture given variance. Happy to further discuss this.

- In line 123, Neurosynth appears to be referred to as a manually curated meta-analytic dataset. Can the authors clarify this? It was my understanding that neurosynth is based on automated abstract parsing.

"Manual curation" corresponds to a selection of maps we previously published. We clarified this point in the methods of the manuscript

" The manual curation consisted in the previously published selection of 590 maps related to specific cognitive processes out of the whole Neurosynth database."

- I'd recommend more details on the structural connectome generation (wrt tractography etc). It's currently hard to know which precise methods were used for preprocessing, tract tracing, and averaging of connectomes across subjects.

These information were reported in previously published papers cited in the manuscript. We now reproduced this information in full in the supplementary material in order to ensure clarity/transparency.

"Structural connectome

Structural connectome data were derived from the diffusion-weighted imaging dataset of 163 participants acquired at 7 Tesla by the Human Connectome Project Team (Vu et al. 2015) (<http://www.humanconnectome.org/study/hcp-young-adult/>) (WU-Minn Consortium; Principal investigators: David Van Essen and Kamil Ugurbil; 1U54MH091657). This was funded by the 16 NIH Institutes and Centers that support the NIH Blueprint for Neuroscience Research, and by the McDonnell Center for Systems Neuroscience at Washington University.

The scanning parameters have previously been described in Vu et al.62. In brief, each diffusion-weighted imaging consisted of a total of 132 near-axial slices acquired with an acceleration factor of 3 (Moeller et al. 2010), isotropic (1.05 mm³) resolution and coverage of

the whole head with a TE of 71.2 ms and with a TR of 7000 ms. At each slice location, diffusion-weighted images were acquired with 65 uniformly distributed gradients in multiple Q-space shells (Caruyer et al. 2013) and 6 images with no diffusion gradient applied. This acquisition was repeated four times with a b-value of 1000 and 2000 s mm⁻² in pairs with left-to-right and right-to-left phase-encoding directions. The default HCP preprocessing pipeline (v3.19.0) was applied to the data (Anderson et al. 2012; Sotiropoulos et al. 2013). In short, the susceptibility-induced off-resonance field was estimated from pairs of images with diffusion gradient applied with distortions going in opposite directions (Anderson et al. 2003) and corrected for the whole diffusion-weighted dataset using TOPUP (Smith et al. 2004). Subsequently, motion and geometrical distortion were corrected using the EDDY tool as implemented in FSL.

ExploreDTI toolbox for Matlab (<http://www.exploredti.com>; Leemans and Jones 2009, Vos et al. 2017) has been used to extract estimates of axonal water fraction (Fieremans et al. 2011). Next, we discarded the volumes with a b-value of 1000 s mm⁻² and whole-brain deterministic tractography was subsequently performed in the native DWI space using StarTrack software (<https://www.mr-startrack.com>). A damped Richardson-Lucy algorithm was applied for spherical deconvolutions (Dell'acqua, F. et al. 2010). A fixed fibre response corresponding to a shape factor of $\alpha = 1.5 \times 10^{-3} \text{ mm}^2 \text{ s}^{-1}$ was adopted, coupled with the geometric damping parameter of 8. Two hundred algorithm iterations were run. The absolute threshold was defined as three times the spherical fibre orientation distribution (FOD) of a grey matter isotropic voxel and the relative threshold as 8% of the maximum amplitude of the FOD (Thiebaut de Schotten, M. et al. 2011). A modified Euler algorithm (Dell'acqua, F. et al. 2013) was used to perform the whole-brain streamline tractography, with an angle threshold of 35°, a step size of 0.5 mm and a minimum streamline length of 15 mm.

We co-registered the structural connectome data to the standard MNI 2 mm space using the following steps: first, whole-brain streamline tractography was converted into streamline density volumes where the intensities corresponded to the number of streamlines crossing each voxel. Second, a study-specific template of streamline density volumes was generated using the Greedy symmetric diffeomorphic normalisation (GreedySyN) pipeline distributed with ANTs (Avants, B. B. et al. 2011). This provided an average template of the streamline density volumes for all subjects. The template was then co-registered with a standard 2 mm MNI152 template using flirt tool implemented in FSL. This step produced a streamline density template in the MNI152 space. Third, individual streamline density volumes were registered to the streamline density template in the MNI152 space template and the same transformation was applied to the individual whole-brain streamline tractography using the trackmath tool distributed with the software package Tract Querier (Wassermann, D. et al. 2016), and to the axonal water fraction maps, using ANTs GreedySyn. This step produced a whole-brain streamline tractography and axonal water fraction maps in the standard MNI152 space.”

Reviewers' Comments:

Reviewer #1:

Remarks to the Author:

The authors have addressed all of my queries and comments in full. I feel this paper will make an excellent addition to the literature.

Reviewer #2:

Remarks to the Author:

The authors have addressed all of my concerns from the previous version of the manuscript.

Signed,

Russ Poldrack

Reviewer #3:

Remarks to the Author:

Overall the authors revised the manuscript appropriately and I would like to thank them for their clarifications. I however felt that some of the previous comments could have been further addressed.

1) With respect to their response to my previous point #3, the authors argue for the absence of an effect of age on the dysconnectome. What my previous comment was mainly alluding to, however, was also whether dysconnectome-derived measures are affected by the composition of the population (with respect to age and clinical signs) that is used to build the normative connectome. Both questions are slightly different, but equally relevant in my opinion.

A) Please acknowledge aging related effects on structural connectome measures in some part of the discussion and how these could contribute to findings that place lesions into normative connectomes based on a particular age range (see eg <https://doi.org/10.1016/j.neuroimage.2017.01.077>).

B) The conclusions of the work could be stronger by replicating the main findings based on dysconnectome findings that were derived from a cohort that is age and sex matched to the stroke population and not just the young adult HCP. Lifespan connectome data with adequate quality should be openly accessible in minimally preprocessed form (e.g. <https://www.humanconnectome.org/study/hcp-lifespan-aging/data-releases>).

C) As mentioned in previous comment, a replication based on a stroke dataset may be desirable as well, but I understand the authors point it may be difficult to come by a diffusion MRI dataset in stroke (note: I only found <https://www.aievolution.com/hbm1901/index.cfm?do=abs.viewAbs&abs=1494> via google, but am not sure the work is already openly accessible). If the authors cannot get a hold on a stroke dwi dataset, the authors may at least want to do A-B and acknowledge the limitation of not showing effects in a stroke connectome in the discussion. The authors mentioned that DWI was used for stroke diagnosis in all patients. Was this DWI sequence not appropriate for tractography?

2) Please briefly discuss pros/cons of using a broad and unselected spectrum of strokes compared to a more narrowly defined cohort of strokes.

3) As a minor point, I would recommend to upload preprocessed diffusion data to a repository that is different from dropbox (e.g. OSF, Dryad, ...).

REVIEWER COMMENTS

Reviewer #1 (Remarks to the Author):

The authors have addressed all of my queries and comments in full. I feel this paper will make an excellent addition to the literature.

Thank you!

Reviewer #2 (Remarks to the Author):

The authors have addressed all of my concerns from the previous version of the manuscript.

Signed,

Russ Poldrack

Thank you

Reviewer #3 (Remarks to the Author):

Overall the authors revised the manuscript appropriately and I would like to thank them for their clarifications.

Thank you for your appreciation

I however felt that some of the previous comments could have been further addressed.

1) With respect to their response to my previous point #3, the authors argue for the absence of an effect of age on the dysconnectome. What my previous comment was mainly alluding to, however, was also whether dysconnectome-derived measures are affected by the composition of the population (with respect to age and clinical signs) that is used to build the normative connectome. Both questions are slightly different, but equally relevant in my opinion.

We apologise for the misunderstanding and now hope that the following revision of the manuscript will be satisfying for you.

A) Please acknowledge aging related effects on structural connectome measures in some part of the discussion and how these could contribute to findings that place lesions into normative connectomes based on a particular age range (see eg <https://doi.org/10.1016/j.neuroimage.2017.01.077>).

Thank you for pointing at this excellent reference that we now cite in the main manuscript as follows

Maintext P. 12 *“Importantly, we privileged the quality of the connectome rather than the age match when deriving the probability of disconnection induced by each lesion. While age-related changes in fractional anisotropy, number of streamlines/trajectories reconstructed, graph theoretical indices have been previously reported in the literature (Damoiseaux et al. 2017), none of these measures were used in our analyses. In the present case, age is not a confounding factor as we previously demonstrated that the shape and spatial extent of tracts (Rojkova et al. 2017), as well as the disconnection estimates (Foulon et al. 2018) are invariant across decades.”*

B) The conclusions of the work could be stronger by replicating the main findings based on dysconnectome findings that were derived from a cohort that is age and sex matched to the stroke population and not just the young adult HCP. Lifespan connectome data with adequate quality should be openly accessible in minimally preprocessed form (e.g. <https://www.humanconnectome.org/study/hcp-lifespan-aging/data-releases>).

Thank you for this suggestion.

We sampled 10 out of the 27 participants of the HCP to create a group matched in age and sex with the stroke dataset.

Despite the spatial resolution of the diffusion of the lifespan human connectome (1.50mm³) project is almost triple the voxel size of the young adult HCP 7T (1.05mm³) we're happy to report that the 'disconnectome maps' had a good reproducibility ($r = 0.866 \pm 0.066$, see supplementary material section E, also reported below for simplicity).

Maintext P. 12 *"In order to confirm this point the disconnection estimates were replicated in a lower resolution age matched sample of 10 participants (Bookheimer et al. 2019; Harms et al. 2018) and indicated a good reproducibility ($r = 0.866 \pm 0.066$; see supplementary material)."*

Supplementary material P. 47-48 ***"E. Replication in an age matched dataset.***

The 'disconnectomes maps' were replicated in a lower resolution age matched sample of 10 participants (average age = 60 with a proportion of male = 0.6).

Structural connectome data were derived from the diffusion-weighted imaging dataset of 10 participants acquired at 3 Tesla by the Human Connectome Project Lifespan study Team^{124,125} (<https://www.humanconnectome.org/lifespan-studies>)

Each diffusion-weighted imaging consisted of a total of 120 near-axial slices acquired with an acceleration factor of 3 (Moeller et al. 2010), isotropic (1.50 mm³) resolution and coverage of the whole head. At each slice location, diffusion-weighted images were acquired with 75 uniformly distributed gradients in multiple Q-space shells (Caruyer et al. 2013) and 5 images with no diffusion gradient applied. This acquisition was repeated four times with a b-value of 1000 and 2500 s mm⁻² in pairs with left-to-right and right-to-left phase-encoding directions. The default HCP preprocessing pipelines (v3.19.0) (Glasser et al. 2013) was applied to the data (Sotiropoulos et al. 2013). Data processing was subsequently performed as described in section A of this supplementary material but for left-to-right and right-to-left phase-encoding directions and ended up with whole-brain streamline tractography in the standard MNI152 space for each subject of this new dataset. The probability of disconnection induced by each of the 1333 lesions was then computed with the 'disconnectome map' tool of the BCBToolkit software using the new processed dataset. Subsequently, similarity between the two set of 'disconnectome map' (original and age matched) was assessed using Pearson correlation. Despite the spatial resolution of the diffusion of the lifespan human connectome (1.50mm³) project being almost triple the voxel size of the young adult HCP 7T (1.05mm³) and the lower number of subjects (10 vs 163 subjects), the two datasets showed a good reproducibility ($r = 0.866 \pm 0.066$)."

C) As mentioned in previous comment, a replication based on a stroke dataset may be desirable as well, but I understand the authors point it may be difficult to come by a diffusion MRI dataset in stroke (note: I only found <https://ww5.aievolution.com/hbm1901/index.cfm?do=abs.viewAbs&abs=1494> via google, but am not sure the work is already openly accessible).

Thank you for pointing at this dataset. Unfortunately, this is not a multidirectional dataset so cannot be used for tractography.

Nevertheless, we would like to insist that most of our analyses were replicated using a split-half approach of our large dataset.

If the authors cannot get a hold on a stroke dwi dataset, the authors may at least want to do A-B and acknowledge the limitation of not showing effects in a stroke connectome in the discussion.

Practically, the acquisition of DTI sequence of sufficient directionality is infeasible in the acute setting because of reduced patient tolerance of long acquisitions and the need to obtain multiple sequences at the same imaging session for clinical purposes. Imaging purely for research purposes is very difficult to justify ethically in the acute setting, and in any event would result in a biased sample, excluding those too ill to tolerate a clinically redundant session. Second, in the presence of a lesion, the tractographic signal will be inevitably disrupted, making it impossible to characterise the white matter architecture in the vicinity of the lesion. This would make estimates of the underlying connectivity patterns substantially worse to an extent likely to obliterate any advantage in closer population matching.

We now acknowledge this point clearly in the main text.

P. 12 *“As we could not directly collect high-resolution diffusion-weighted imaging from each patient due to limited clinical settings, instead, lesions were placed in healthy connectomes to estimate disconnections (Thiebaut de Schotten et al. 2015; Foulon et al. 2018; Theiabut de Schotten et al. 2018; Pacella et al. 2019; Dalla Barba et al. 2018; Salvalaggio et al. 2020).”*

The authors mentioned that DWI was used for stroke diagnosis in all patients. Was this DWI sequence not appropriate for tractography?

Clinical DWI sequences applied in the acute setting are not designed for tractography, and to our knowledge no sufficiently rapid sequence is in widespread use.

2) Please briefly discuss pros/cons of using a broad and unselected spectrum of strokes compared to a more narrowly defined cohort of strokes.

We now added this point in the text accordingly.

P. 13 “Lesion data was derived from 1333 patients admitted between 2001 and 2014 to University College London Hospitals (UCLH) with a clinical diagnosis of acute ischaemic stroke confirmed by diffusion-weighted imaging (DWI). Since DWI was routinely performed on the majority of attending patients, the sample was representative of the population, constrained mostly by contraindications and tolerability. The advantage of using a broad and unselected spectrum of strokes compared to a more narrowly defined cohort of stroke is that it is clinically relevant and generates characteristic patterns of disconnection that are the main variable of interest of our study. In contrast, narrowly defined cohorts of stroke still suffer from a biased distribution and are not representative of the stroke population and consequently cannot be used for data-driven clinical predictions”

3) As a minor point, I would recommend to upload preprocessed diffusion data to a repository that is different from dropbox (e.g. OSF, Dryad,).

Thank you for the suggested repository. This is great! We now also uploaded the dataset on OSF as per your request and provided the link P. 20

Reviewers' Comments:

Reviewer #3:

Remarks to the Author:

The authors have addressed my comments and i would like to congratulate them for their work.